# Stress, Allostatic Load, and Neuroinflammation: Implications for Racial and Socioeconomic Health Disparities in Glaucoma

**DOI:** 10.3390/ijms25031653

**Published:** 2024-01-29

**Authors:** Colleen E. McDermott, Rebecca J. Salowe, Isabel Di Rosa, Joan M. O’Brien

**Affiliations:** 1Department of Surgery, University of Utah, Salt Lake City, UT 84101, USA; 2Scheie Eye Institute, Perelman School of Medicine, University of Pennsylvania, Philadelphia, PA 19104, USA; rebecca.salowe@pennmedicine.upenn.edu (R.J.S.); isabel.dirosa@pennmedicine.upenn.edu (I.D.R.);

**Keywords:** glaucoma, neuroinflammation, vascular homeostasis, HPA axis, neuroendocrine signaling, allostatic load, chronic stress, socioeconomic disparities, racial disparities

## Abstract

Glaucoma is the leading cause of irreversible blindness, and its pathophysiology includes neuroinflammatory changes. The present therapies for glaucoma target pressure-lowering mechanisms with limited success, making neuroinflammation a target for future interventions. This review summarizes the neuroinflammatory pathways seen in glaucoma and their interplay with stress. Glucocorticoids have been shown to activate proinflammatory glial cells, contributing to the neuroinflammation in glaucoma. Glucocorticoids have also been shown to increase the IOP directly. Stress-associated autonomic dysfunction can affect the vascular homeostasis in the retina and create oxidative stress. Diabetes, hyperglycemic-mediated endothelial damage, and vascular inflammation also play important roles in the neuroinflammation in glaucoma and diabetic retinopathy. Psychosocial stress has been implicated in an increased IOP and glaucoma outcomes. People who experience maladaptive chronic stress suffer from a condition known as allostatic load, which describes pathologic neuroendocrine dysregulation. The effects of allostatic load and chronic stress have been studied in patients affected by a lower socioeconomic status (SES) and marginalized racial identities. A lower SES is associated with higher rates of glaucoma and also affects the access to care and screening. Additionally, people of African ancestry are disproportionately affected by glaucoma for reasons that are multifactorial. In conclusion, this review explores neuroinflammation in glaucoma, highlighting opportunities for future investigation.

## 1. Introduction

As the leading cause of irreversible blindness, glaucoma is a neurodegenerative disease that poses a major vision threat to patients worldwide, with nearly 100 million people estimated to be affected by 2040 [1]. The pathophysiology of glaucoma includes a complex interaction between an elevated intraocular pressure (IOP) and subsequent neuroinflammatory and neurodegenerative changes in the retina. IOP elevations have been shown to stimulate a proinflammatory environment in the retina, leading to retinal ganglion cell (RGC) death and progressive vision loss [1].

Though glaucoma is characterized by the death of RGCs and their axons, no existing treatments target this cell type. Presently, all therapies for glaucoma target one mechanism of the disease: the elevated IOP. However, nearly 40% of patients experience disease progression despite IOP-lowering interventions [2]. Additionally, a subset of patients experience the progressive degradation of RGCs despite a normal IOP; this condition is defined as normal-tension glaucoma (NTG) [3]. These findings suggest that there are additional non-IOP-related mechanisms in glaucoma that, if elucidated, could represent future targets for therapy. There is likely a significant contribution of genetic risk factors to glaucoma pathogenesis as well, with the heritability (h2) estimated between 0.17 and 0.81 [4]. The risk loci identified for glaucoma may provide insight into glaucoma pathogenesis.

Chronic and systemic inflammation have been implicated in glaucoma and in other neurodegenerative diseases, such as Alzheimer’s disease (AD). Some genes that have been implicated in the development of Alzheimer’s disease have also been implicated in glaucoma, such as *APP* and *MAPT* [5]. However, unlike other neurodegenerative diseases, the retinal microenvironment in glaucoma differs from the rest of the central nervous system (CNS), as it is a relatively immune-privileged site [3,6,7]. Despite this, one of the major genes implicated in the risk of glaucoma is *TBK1*, which is involved in the innate immune response, suggesting inflammatory and immunomodulatory components to the disease pathogenesis [8].

Neuroprotection and the mitigation of neuroinflammation are gaining relevance as avenues for the treatment of glaucoma, although the mechanisms of neurodegeneration in glaucoma are still being elucidated [9]. In a 2022 article, Arrigo et al. reviewed the body of work exploring how the scope of glaucoma extends beyond the retina and optic nerve injury, suggesting that this disease represents a more complicated neurodegenerative disorder [9]. Importantly, neurodegenerative changes seen in glaucoma are not confined to the central visual pathway, but are seen in areas of the brain involved in emotional and physiological responses to stress as well [10].

The purpose of this review was to examine the role of chronic neuroinflammation in the development and progression of glaucoma. This will be explored in the context of chronic stress as a mediator and contributory factor to neuroinflammatory processes and RGC degeneration. We will explore how stress can affect these cellular processes and play a role in the development and progression of glaucoma. We will then review how stress may affect individuals on a molecular, patient, and community level. This will ultimately be tied in to the major health disparities observed in glaucoma.

## 2. Neuroinflammation in the Retinal Microenvironment

The retinal immune microenvironment is strictly regulated by glial cells, such as astroglia and microglia. These cells regulate the immune responses and inflammatory reactions to threats [3]. Astrocytes interact closely with the blood–brain barrier (BBB), and therefore play an important role in sensing systemic inflammation or pathogen-related peptides and regulating immune factors within the CNS [11]. Microglial cells are major immune cells of the CNS. They secrete numerous cytokines and growth factors that can influence the cellular environment, and they interact closely with astrocytes. While microglia function in a protective capacity against an acute pathogenic insult, they can also become dysregulated, leading to chronic neuroinflammation. There are two distinct phenotypes for microglia. The M1 phenotype secretes proinflammatory cytokines, and uncontrolled activation can lead to inflammatory-mediated neurotoxicity [3]. These cells also play a role in modulating and resolving neuroinflammation [12]. Conversely, the M2-phenotype cells produce anti-inflammatory factors [3]. Microglial dysfunction has been implicated in the development of psychiatric and neurodegenerative diseases, including glaucoma [6]. There has been extensive research on particular glial cell interactions in neuroinflammation and glaucoma, and while these nuanced mechanisms are outside the scope of this review, they are summarized in a 2021 review by Zhao et al. [13].

## 3. Glucocorticoids and Glial Cell Interactions in Neuroinflammation

As previously discussed, there are extensive molecular interactions involved in the neuroinflammatory pathways underlying glaucoma. One proposed pathway involved in glial cell dysfunction and neuroinflammation is the interaction between glial cells and glucocorticoids. Although glucocorticoids are often used as potent anti-inflammatory agents, there is evidence that even an endogenous exposure to glucocorticoids may be implicated in glaucoma development [14].

Although the precise link between glucocorticoid exposure and glaucoma development is an area of ongoing research, one proposed pathway is discussed by Pereira et al. This paper describes a self-reinforcing cycle between the hypothalamic–pituitary–adrenal (HPA) axis and microglial cell activation [6]. Stress has been shown to cause the dysregulation of microglial cells, playing an important role in neuroinflammation and glaucoma. This stress-induced dysregulation directly affects microglial cells and promotes the release of proinflammatory cytokines. The HPA axis is also stimulated in response to stressors, which causes the downstream release of glucocorticoids [6]. The authors postulated that neuroinflammation triggered by activated microglial cells can directly upregulate glucocorticoid release from the HPA axis, which can in turn interact with glucocorticoid receptors on the microglia, promoting microglial proliferation [6]. The continuous activation of the HPA axis can result in diminished feedback and the constitutive release of glucocorticoids [15]. In a 2021 review, Picard et al., suggested that increased glucocorticoid signaling can push microglia into a proinflammatory (M1) state, which then creates an exaggerated response to secondary pathogen insults [15]. This suggests that glucocorticoid effects on glial cells may play a role in glaucoma development.

In addition to glucocorticoid-mediated effects on glial cells, chronic stress has been shown to exacerbate neuroinflammation by causing disruptions to the BBB, allowing systemic proinflammatory mediators to enter the CNS [15]. Additionally, there is emerging evidence that other hormones secreted by the HPA pathway, such as noradrenaline, can modulate microglial cell activity as well. This noradrenaline-mediated effect provides an alternative mechanism for stress-associated glial cell dysfunction [16].

The mechanisms described here pose an explanation for how HPA-axis dysregulation can influence neuroinflammation, and ongoing research is needed to further identify the possible molecular targets for glaucoma therapies. One potential avenue for novel therapies related to stress and neuroinflammation is the suppression of the proinflammatory or M1 phenotype of microglial cells. In a 2019 animal model of chronic unpredictable mild stress, minocycline was found to ameliorate neuro-immune activation and reverse the M1 response [17]. While a full review of other potential targets is beyond the scope of this review, the comprehensive review of potential immunomodulatory targets for glaucoma by Bell et al. can be referred to for additional information [18]. In general, the interaction between the immune system, HPA dysregulation, glucocorticoid signaling, and glaucoma development presents an exciting avenue for glaucoma therapy that goes beyond lowering the IOP. Additional insight into therapeutic opportunities may be gleaned from exploring the causes of chronic HPA disruption.

## 4. Glucocorticoids and Intraocular Pressure (IOP)

A well-described phenomenon of iatrogenic or glucocorticoid-induced glaucoma (GIG) links the exogenous administration of corticosteroids to ocular hypertension. This was first described in the 1950s, and has become an important interest, both for the avoidance of the iatrogenic causes of optic nerve damage and for providing insights into the pathophysiology of glaucoma [19]. A recent review by Patel et al. explored the possible pathophysiologic mechanisms in GIG. The prevailing theory is that glucocorticoids induce multifactorial damage to the trabecular meshwork that inhibits the drainage of the aqueous humor and causes an increased IOP [14]. There are two glucocorticoid receptor (GR) variants, known as GRα and GRβ. The role of GRβ has been implicated in downregulating the activity of GRα and blunting the response to glucocorticoids [14]. In eyes isolated from patients with primary open-angle glaucoma (POAG), decreased GRβ activity was observed, suggesting that there is a heightened glucocorticoid responsiveness in POAG patients [20]. Additionally, about 40% of the general population exhibited an increased IOP in response to exogenous steroids, compared to nearly 100% of POAG patients [14,21,22]. There is also evidence that individuals who demonstrate an increased IOP when exposed to exogenous steroids may be at an increased risk for POAG in the future [14]. Although the mechanism for this is not completely understood, in a review of steroid-induced glaucoma (SIG), Roberti et al., hypothesized that genetic variations that lead to a differing expression of GRα and GRβ may play a role in different response phenotypes [23].

In general, genetic variants may play a role in aqueous humor drainage via the TM function. *MYOC*, the gene that encodes myocilin, is a genetic risk factor for glaucoma. Myocilin is thought to play a role in extracellular matrix remodeling in the TM, although this is an area of ongoing research [8]. This alteration in TM functionality and impaired aqueous humor drainage may be exacerbated by glucocorticoid-mediated TM dysfunctions. Additional research is needed to identify specific genetic variants that may play a role in SIG.

There are multiple routes by which patients may be exposed to glucocorticoids. Both systemic applications (i.e., oral, intravenous, inhaled, or percutaneous) and local applications (i.e., topical, intravitreal injection, or intravitreal implant) of steroid exposure have been explored. In a major 2020 review by Roberti et al., the authors summarized the evidence of an increased ocular pressure in patients who had been treated with steroids. The exact percentages of patients who experienced increases in their IOP varied from study to study, with a wide range in the degree of IOP increases across all forms of steroid administration. The authors emphasized that topical and local steroids may cause patients to present with an increased IOP after days to weeks, while systemic steroids may lead to an elevated IOP even years later. There is no definitive evidence that one route of administration causes more severe IOP elevations or increases the glaucoma risk compared to other routes [23]. Because patients are given steroids via various routes for numerous clinical conditions, a head-to-head comparison of the routes of steroid administration on glaucoma development is difficult. The key finding from the review by Roberti et al. was that all patients with prior steroid exposure are at risk for IOP elevation and glaucoma, and these patients should be screened appropriately [23].

In addition to the effects of exogenous steroids on the IOP, there is also evidence to suggest that endogenous glucocorticoids may affect the IOP. The naturally-occurring diurnal variation in cortisol levels mirrors the diurnal fluctuations in the IOP. There is also a lack of diurnal variation in the IOP in patients who have undergone bilateral adrenalectomies [24]. Patel et al., noted that POAG patients have been shown to have higher levels of serum cortisol and cortisol found in the aqueous humor. Although this area is in need of further research, there is also evidence of aberrant cortisol signaling in POAG patients [14]. This suggests that, in addition to the glucocorticoid-mediated contribution to neuroinflammation discussed above, endogenous and exogenous glucocorticoid signaling may also play a role in the IOP elevation in glaucoma development.

## 5. Stress, HPA Dysregulation, and Energy Homeostasis in the Retina

As discussed above, there are several molecular mechanisms through which glucocorticoids can exacerbate both the neuroinflammatory and IOP-elevation components of glaucoma. This, then, suggests that chronic stress, or other conditions that activate the HPA axis, may play a role in glaucoma development.

### 5.1. Stress-Induced Mitochondrial Dysfunction

Another avenue where stress may play a role in glaucoma development is that of stress-induced mitochondrial dysfunction. RGCs, which are a group of neurons that connect the retina to the brain, are one of the most metabolically active tissues in the body, with extremely high energy requirements, and therefore, they are very sensitive to changes in energy expenditure [2]. Brain mitochondria play an integral role in the body’s response to stress. They adapt to the metabolic needs during periods of stress via glucocorticoid signaling [25]. However, chronic stress causes the dysregulation of mitochondrial pathways, which Morella et al., described as “mitochondrial allostatic load” [25]. This weathering of the brain mitochondria is linked to structural and functional changes in mitochondria that cause oxidative stress and inflammation [25]. In a review, Duarte et al. examined the complex interactions between dysregulated mitochondria and the immune system, indicating a multifactorial relationship. The author hypothesized that mitochondrial-damage-associated molecular patterns (DAMPs) create and perpetuate a cycle of inflammation [26]. This suggests that mitochondrial damage due to stress and allostatic weathering may not only be a causative derangement, but may also perpetuate retinal energy dysregulation. This resulting energy imbalance was implicated as an important pathway that relates chronic weathering due to stress and the development of glaucoma in a review by Dada et al. [27].

### 5.2. Neurovascular Dysregulation

While neuroinflammation and inefficient mitochondrial energy generation are two possible mechanisms for injury to RGCs in glaucoma, there is likely neurovascular involvement as well. Astrocytes, which are implicated in neuroinflammation pathways through interactions with microglia, also play an important role in regulating the circulation within the brain and altering the vascular tone according to the brain energy needs [28]. This suggests that chronic neuroinflammation damages the brain mitochondria’s ability to provide energy for highly metabolically active RGCs. It also suggests that neuroinflammation may impede the brain’s ability to autoregulate the blood supply to RGCs to meet the tissue oxygen demands. Catecholamines released in response to stress can also trigger systemic vasoconstriction, which further decreases the blood supply to the optic nerve. This can lead to optic neuropathy without associated changes in the IOP, as noted in cases of NTG [27].

There has been increased interest in exploring the relationship between NTG and cerebrovascular disease. Some studies have shown an association between a greater cerebral infarct burden and white matter lesions in patients with NTG as well as POAG. These have been summarized in a review by Nucci et al. [29]. A recent study by Zhang et al., also implicated the vasoreactivity of the posterior cerebral artery (PCA) in response to visual field stimulation as a harbinger of future visual field loss [30]. In a 2012 case-control study, NTG patients showed evidence of cardiac autonomic dysfunction during a 24 h measurement. They were found to have an altered heart rate variability associated with increased sympathetic activity, although there was no difference in the blood pressure variation compared to controls [31]. 

In the 1990s, a concept pioneered by Josef Flammer, known as Flammer syndrome, was described. This syndrome is characterized by vascular dysregulation in several areas of the body, including the eye. The nature of vascular regulatory abnormalities is not well understood, but the dysregulation of the autonomic nervous system seen in chronic stress may play a role. There is ongoing research into the role that Flammer syndrome may play in the pathogenesis of NTG [32]. A 2017 study found a higher association of Flammer syndrome symptoms in NTG patients compared to controls [33]. Additionally, a 2022 review emphasized that disruptions to ocular blood flow (OBF) may be associated with astrocyte activation and oxidative stress in NTG patients. The authors emphasized that the role of OBF is an ongoing area of research and that it is important to look beyond IOP-lowering interventions moving forward [34].

Although the mechanism of RGC loss in glaucoma is still an ongoing area of research, this ischemic insult due to mitochondrial dysfunction and poor autoregulation could further exacerbate RGC demise and lead to the worsening of glaucoma. As discussed above, NTG patients without Flammer syndrome were found to have autonomic dysfunction and increased sympathetic activity [32]. This suggests that stress, HPA activation, and autonomic dysregulation may contribute to glaucoma progression via these neurovascular pathways.

## 6. Diabetes Mellitus (DM), Vascular Inflammation, Diabetic Retinopathy (DR), and Glaucoma

### 6.1. Vascular Inflammation as a Common Pathway for Degenerative Retinal Diseases

In addition to the autonomic dysregulation described above, there are additional complexities of the retinal vascular microenvironment that may be associated with glaucoma pathogenesis. There is a strong association of DM with glaucoma. A meta-analysis that included over 2.7 million patients by Al Darrab et al. found that the duration of DM and elevated fasting glucose levels were strong risk factors for DR as well as independent risk factors for an elevated IOP [35]. A population cross-sectional study of over 43,000 patients by Mahmood et al., also found that an A1c > 6.4 increased the risk of glaucoma development [36].

One hypothesis for the connection between DM and glaucoma is that endothelial cell dysfunction caused by chronic hyperglycemia can lead to vascular inflammation [37]. A 2019 review by Soto et al., described in detail the complex interplay between vascular inflammation and the progression of major retinal diseases, including DR and glaucoma [37]. Vascular inflammation is considered the chief driver of the development of diabetic retinopathy [38]. This vascular inflammation in the retina can lead to the infiltration of microglia, perpetuating neuroinflammation in the retinal microenvironment. The authors emphasized that DR and glaucoma likely share several common pathways that lead to RGC degeneration, including microvascular inflammation and the perpetuation of microglial activation and neuroinflammation [37].

### 6.2. Hyperglycemia and Stress

The effects of diabetes on glaucoma progression are likely multifaceted; in addition to the vascular inflammation and neuroinflammatory pathways described above, stress, HPA dysfunction, and glucocorticoid signaling may also play a role. As part of its physiologic function within the body, endogenous glucocorticoid release prepares the body for “fight or flight” by increasing the serum glucose levels [39]. This process is often associated with worsening glycemic control in diabetic patients. However, there is evidence that early life stress and HPA dysregulation can be associated with poor long-term metabolic health outcomes [40]. A German population study recently linked perceived chronic stress levels with an increased risk of type 2 DM [41]. While this association is likely complex, it brings to light an important way that stress may impact glaucoma development. As HPA dysregulation and chronic stress can lead to glucocorticoid-mediated hyperglycemia or the development of DM, this process can then lead to further damage of the RGCs via the vascular inflammatory pathways described above.

## 7. Psychosocial Stress, IOP, and Neuroinflammation

### 7.1. Relationship between Stress and Increases in IOP

In addition to the microvascular sequelae mediated by hyperglycemia and DM, the relationship between psychosocial stress and glaucoma progression is multifactorial and is still an area of ongoing study. In addition to the implications discussed above for neuroinflammation and the molecular mechanisms of glucocorticoid signaling, there is a question of how stress directly affects the IOP in vivo. 

In a 2020 case control study, a group of healthy subjects had their IOP changes measured after stress induced by the Trier social stress test (TSST). Diurnal tension curves were created for each subject a week before the intervention. A mean elevation of >1 mmHg in each eye was demonstrated in the experimental group. These subjects also demonstrated significant elevations in their salivary cortisol and heart rate. The study was small (N = 28), but it suggested that stress directly influences the IOP in real time in human subjects [42]. This may be via a glucocorticoid-mediated response seen in the exogenous steroid-induced IOP elevation described earlier; however, due to the short time period between the stressor and the effect, it may be more directly related to blood pressure dynamics in response to sympathetic activation. 

Lee et al., assessed the relationship between stress, the IOP, and the retinal nerve fiber layer (RNFL) thickness in a 2020 study where biometric data were collected from healthy volunteers who underwent the TSST. These investigators used the RNFL thickness as a measure of the integrity of RGCs. They found that there were no differences between the TSST response types and the IOP or RNFL thickness; however, they found that there was an inverse relationship between the baseline endogenous adrenocorticotropic hormone (ACTH) levels and the RNFL thickness (*p* = 0.009 globally). These changes were small and likely subclinical with respect to visual field changes, but the authors postulated that this effect may become more pronounced over a person’s life course, as the volunteers were in their late teens and early twenties [43].

### 7.2. Psychosocial Stress and Neuroinflammation

While the studies above explored the effects of acute stressors, it is also important to evaluate the role of chronic psychosocial stress, and how it may relate to neuroinflammation and glaucoma. First, we will discuss the background of how chronic social stressors have been implicated in neuroinflammation.

A relationship between social rejection, inflammation, and the development of depression has emerged in studies regarding another neuroinflammatory condition, major depressive disorder. A nascent theory for this relationship has been described by Slavich et al. as the social signal transduction theory. This hypothesis suggests that social threats and other forms of chronic adversity lead to a proinflammatory state that can contribute to the pathogenesis of MDD via neuroinflammation [44]. This theory operates on a similar premise to McEwen’s concept of allostatic load (discussed in detail in the subsequent sections), and takes into account how chronic social adversity can cause biological changes in the body’s homeostatic mechanisms. In particular, Slavich and Erwin describe how adversity can sensitize the CNS to promote a proinflammatory environment. They caution that this can lead to a sustained perception of threat in the body, which can cause hypervigilance and anxiety in the short term and disrupted sleep or social withdrawal in the medium term, and can eventually lead to inflammatory and neurodegenerative diseases in the long term. They indicate that these pathways are likely mediated by systemic inflammatory markers such as IL-1, IL-6, and TNF-α [44]. Stress, in general, is a stimulatory factor of proinflammatory cytokines such as IL-6 and TNF-α [10]. This also represents a similar cytokine mechanism to that described in the allostatic load theory (discussed below). This premise was illustrated in human subjects in a study by Murphy et al. in 2012, which demonstrated the upregulation of inflammatory gene expression in adolescents exposed to targeted social rejection by their peers [45].

Additionally, a 2017 review article by Weber et al. described the complex interactions between chronic stressors and communication between the innate immune system and the brain. They described neuroinflammation as a cause of the trafficking of peripheral monocytes to the CNS, which is facilitated by an impaired BBB. These monocytes then communicate with microglia to create a pro-inflammatory environment [46].

### 7.3. Stress, Neuroinflammation, and Glaucoma

Building on the above evidence of the molecular hallmarks of neuroinflammation seen in chronic stress, these inflammatory cytokines were further implicated in glaucoma in a study conducted by Cvenkel et al. [47]. The authors found that eyes with trabeculectomy failure showed significantly higher levels of these inflammatory mediators in the aqueous humor than eyes with a successful intervention. The authors suggest that this proinflammatory environment may contribute to the progression of the disease and worse outcomes after glaucoma surgery [47].

On a more macroscopic scale, chronic social stress has been implicated in glaucoma development and progression. In a cross-sectional study by Ji et al., a cohort of glaucoma patients were stratified by stress level according to their responses on the perceived stress scale. They found that the high-stress group had more severe visual field defects and lower best-corrected visual acuity than the low-stress group. The authors discussed that this interaction between stress and a worsening disease could be multifactorial in nature. They cited autonomic dysregulation and vascular autoregulation dysfunction (as discussed above) as plausible mechanisms for the disease progression. However, they also discussed that mental stress may play a detrimental role in treatment adherence and noncompliance. A vicious stress cycle could exist in which stress exacerbates glaucomatous progression, causing a decrease in visual acuity, which heightens stress [48].

In a study on patients with NTG, the patients were assessed for the presence of emotional and neuropsychiatric complaints. The NTG subjects were found to have more psychosomatic discomfort than the control group and were found to be more emotionally unstable. The authors emphasized that this suggests that there is psychosocial stress involved in NTG, but they questioned whether this is a cause or an effect [49]. The mitochondrial dysfunction and aberrant vascular regulation described above could play a role in this relationship.

In a 2018 review, Sabel et al. acknowledged that there is a clear relationship between vision loss and subsequent stress, but also made an argument for a causative relationship between stress and vision loss, implicating many of the same pathways cited in this present review [10]. This represents an area that is incompletely understood and would be amenable to further research. The mitigation of psychosocial stress could also serve as an opportunity for therapeutic intervention. 

## 8. HPA-Axis Disruption and the Allostatic Load Hypothesis

As mentioned briefly above, there exists a continuum of stress that the body experiences, from acute stressors to chronic psychosocial stressors to pathological stress, that is associated with dysregulated autonomic nervous system function, known as allostatic load. Coined by McEwen et al., in the 1990s, the concept of allostatic load describes the effects of chronic patient- and community-level stressors over a person’s life course on the body’s ability to maintain homeostasis. McEwen et al., cautioned that, while neuro-endocrine and neuroinflammatory stress responses are often appropriate and allow the body to withstand stressors intermittently, chronic and frequent activation can lead to the dysregulation of these systems [50]. As a result, individuals who experience sustained adversity create a biological memory of these multiple threats, resulting in the long-term dysregulation of homeostasis. It can be thought of as cumulative “wear and tear” on the body, brought on by chronic adversity [51]. These dysregulated neuroendocrine and inflammatory responses have been implicated in adverse health outcomes such as diabetes, obesity, atherosclerosis, depression, Alzheimer’s disease, and cognitive issues [52,53]. As discussed above, diabetes and metabolic disease play a role in the progression of glaucoma and other retinal diseases, and the impact of allostatic load on perpetuating poor metabolic health may worsen retinal diseases. Additionally, allostatic load has been directly implicated in glaucoma pathogenesis, and this will be described in more detail below [27].

In the well-described Trier social stress test (TSST), which includes public speaking and arithmetic challenges, individuals are categorized into three phenotypes: anticipatory responders, reactive responders, and non-responders, based on cortisol changes in response to the stressor [43]. Non-responders are considered to have a pathological dysregulation or blunting of the stress response that has been linked to early life adversity and stressful childhood events [54]. This dysregulated stress response has been shown to be associated with an increased risk for obesity, substance use disorders, MDD, eating disorders, and poor neurocognitive function [54]. In addition to adverse childhood events and stressful life events, socioeconomic inequality has been noted to be an important contributor to allostatic load, as have racial discrimination, emotional trauma, and social alienation [53].

In a 2020 review by Dada et al., the authors created a multifactorial framework implicating allostatic load in glaucoma. There are three major pathways by which allostatic load is thought to play a role in glaucoma development: (1) cortisol effects on the IOP, leading to classical high-tension glaucoma; (2) an energy imbalance and ischemic insult at the optic nerve due to impaired vascular regulation, as seen in NTG; and (3) the upregulation of inflammatory cytokines, such as IL-6 and TNFα, that can perpetuate neuroinflammation [27]. There is a formalized allostatic load (AL) score that creates a cumulative assessment of allostatic weathering and the effects of stress. It is defined by various neuroendocrine, cardiovascular, and inflammatory biomarkers, including blood pressure, body mass index (BMI), glycosylated hemoglobin (Hgb A1c), total cholesterol, triglycerides, albumin, C-reactive protein (CRP), homocysteine, and creatine clearance [51]. In an analysis of the National Health and Nutrition Examination Survey (NHANES) cohort, the relationship between visual acuity, allostatic load, and mortality was assessed. A visual acuity impairment itself was found to be associated with increased allostatic load in a multivariate regression analysis (*p* = 0.01). The previously described effect of visual acuity on mortality was also shown to be significantly mediated by the allostatic load score [51]. This suggests that allostatic load may be both a contributing factor to the development of glaucoma and perpetuated by the stress and anxiety associated with the progressive visual impairment in this disease.

## 9. Socioeconomic Status (SES), Glaucoma, and Neuroinflammation

An exploration of the contributors to chronic stress, such as the SES, can help provide further information on the effect of allostatic load on glaucoma. Glaucoma is often asymptomatic in the early stages, making early detection critical for the mitigation of the disease effects [55]. Unfortunately, a lower SES has been associated with a diagnosis of end-stage glaucoma, and a disproportionate number of patients with POAG are of a lower SES [55]. The influence of socioeconomic deprivation and community-level disadvantages has been implicated in pathological changes due to chronic stress, particularly in the allostatic load model. There have been a number of studies that have examined the SES and its implications in the risk for the development or progression of glaucoma. In a study using the Korean National Health and Nutrition Examination Survey (KNHANES), a multivariate analysis of >24,000 patients demonstrated that the odds of glaucoma decreased as the income level increased, although they also observed an increase at the highest income quartile [56]. 

As a proxy for the SES, a machine learning model by Nusinovici et al. demonstrated that a lower education level is a risk factor for primary angle closure glaucoma (PACG), as individuals with no formal education or only elementary education were shown to be at a higher risk for the disease. The authors suggested that this could be due to relative differences in health literacy in this population, which could affect their healthcare-seeking behavior [57].

In a study comparing gender differences in the global glaucoma disease burden, Ye et al. found that a greater gender disparity was seen in countries with lower ratings on a national socio-economic metric, the human development index (HDI). This metric is a summary of the countrywide income, education, and health. The authors emphasized that, in lower-HDI countries, there may also be a need for increased access to affordable eye care [58]. Similarly, when assessing the global glaucoma burden, Wu et al. found that a lower SES and a lower mean number of years of schooling were associated with a higher glaucoma burden, as measured using disability-adjusted life years (DALYs) [59]. This finding is likely multifactorial, as insurance issues, transportation issues, and health literacy (particularly with respect to the importance of preventative eye care) may play contributory roles [55]. Other multi-level factors in the global glaucoma burden, as discussed in Ye et al.’s review, include geographic challenges in access to care, low education levels, transportation difficulties, and lower average numbers of ophthalmologists in areas with less economic development [58].

These studies suggest that there is evidence that the SES plays a factor in a delay in diagnoses, adherence to care, and access to care, but could it also play a role in the development and progression of glaucoma directly? In a review article, Olude et al. hypothesized that a damaging cycle created by poverty leading to malnutrition could leave individuals susceptible to neuroinflammation and sequelae, such as neurodegenerative disorders and neurocognitive impairment, which could then perpetuate poverty [12].

There is certainly evidence that individuals who are affected by socioeconomic deprivation may be at risk for chronic stress, an increased allostatic load burden, and HPA dysregulation. The implications of these derangements on neuroinflammation may also play a role in glaucoma development and progression. A review by Szanton et al. emphasized that a lower SES is associated with a higher allostatic load burden [60]. Although more research on this topic needs to be conducted, this framework could provide an avenue for future study to elucidate the true burden of socioeconomic depravity on health disparities that is not accounted for by downstream factors (e.g., access to healthcare). 

## 10. Racial and Ethnic Disparities

In addition to the effects that factors such as the SES can have on glaucoma disparities, there are also well-described disparities between racial and ethnic groups that may be exacerbated by chronic stress. Glaucoma disproportionately affects individuals of African ancestry, both with respect to its prevalence and severity. Despite this disparate incidence, this population has been found to be underrepresented in glaucoma research and clinical trials [61,62]. Efforts have been made to recruit patients of African ancestry to participate in research, with the hopes of elucidating why glaucoma disproportionately affects this ancestral group [62]. This disparity is likely multifactorial in origin, but it is possible that the connection between chronic stressors and neuroinflammation may play a role in explaining this disparity. 

Patients of African ancestry have been shown to be diagnosed with glaucoma in later stages, which may be related to a disparate access and interaction with outpatient preventative eye care [55]. In a study of nearly 80,000 Medicare beneficiaries, the eye care differences among Medicare-defined racial categories persisted, despite a correction for the SES; Black/African American patients utilized less outpatient and preventative care, but more commonly underwent surgical procedures and were seen more commonly in emergency or inpatient settings [63]. Medicare-defined Hispanic patients were also shown to be less likely to have an outpatient follow-up or to receive outpatient eye examinations. The authors discussed reasons for why race and ethnicity persist as risk factors for the underutilization of eye care, after correcting for the SES. They emphasized that there are well-documented racial/ethnic disparities in access to healthcare. These are linked to both structural racism and inter-generational disparities across multiple sectors such as housing, education, finance, and political representation [63]. Additionally, experiences with past racial discrimination in medical research and in the healthcare system has been observed as a factor that contributes to the decisions of individuals of African ancestry on whether to participate in medical research [62]. This likely also plays a contributory role in their interactions with the healthcare system for regular medical care. 

Racial disparities in the access to care and involvement in research can exacerbate existing cases of glaucoma, yet, despite less usage of outpatient screening, individuals of African ancestry make up a disproportionately large portion of the patients affected by glaucoma [64]. An extensive review into the major risk factors for POAG in individuals of African ancestry conducted in 2015 identified several common risk factors in the general population, including an older age, the male gender, a positive family history, and an elevated IOP [65]. Several anatomical factors, such as a thinner central corneal thickness, vascular abnormalities, myopia, and decreased corneal hysteresis, were also identified as risk factors. The authors recommended further genetic analyses and the exploration of pathologic mechanisms in this population, and ongoing work is being conducted to further elucidate the causal mechanisms of glaucoma development in individuals of African ancestry [65].

The genetic risk factors that have been identified in populations of European ancestry may have only limited relevance to populations of African ancestry [66], necessitating further research into the genetic components of POAG in the over-affected individuals of African ancestry. In a large genome-wide association study of three large cohorts of African ancestry, including a POAAGG cohort, the researchers discovered three likely causal variants for POAG [67].

In a study by Siesky et al., the authors found that, in patients of African descent, there was a significantly larger increase in the avascular area of the retina compared to patients of European descent. This was correlated with a change in the optic nerve and retinal morphology [68]. This observation suggests that there are more complex factors involved in the susceptibility to vascular ischemia in the African-ancestry population that require further research to elucidate. These findings collectively suggest that some individuals of African ancestry may be genetically predisposed to be adversely affected by neuroinflammatory processes, which could place these patients at a greater risk for glaucoma development and progression. 

Additionally, the social and multi-level determinants of health that are experienced by African-ancestry communities may contribute to neuroinflammation. In a 2022 review article, Olude et al. explored stress-induced neuroinflammation on the continent of Africa. They cited numerous possible contributors to neuroinflammation, including poverty, malnutrition, pollution, early life stress, violence, and infectious outbreaks. The authors cautioned that the 1.3 billion people living on the continent of Africa who are exposed to these stressors may see an increasing incidence of neuroinflammatory disorders [12]. There is additional evidence implicating systemic social factors on the health outcomes in African-ancestry patients beyond neuroinflammation. One possible pathway connecting these social determinants of health to disease pathogenesis in the African-ancestry population is via the allostatic load mechanism. A small biometric study on equal-SES African American and White participants demonstrated an association of race with a higher allostatic load score [68]. In the aforementioned 2014 NHANES cohort study, a multivariate regression analysis demonstrated that a self-reported non-Hispanic Black ethnicity was a risk factor for elevated allostatic load, as determined using a biometric analysis (*p* < 0.01) [51]. A 2020 analysis of the US biomarker project cohort examined the experiences of racial discrimination in participants using three independent, validated assessment tools. Participants who identified as Black were found to have significantly higher pervasive discrimination scores as well as allostatic load scores. The associations between discrimination and allostatic load, however, did not differ by race in a significant way. This study suggested that, while more research needs to be conducted, there may be evidence that pervasive racial discrimination is associated with greater physiological dysfunction, and likely serves as a chronic stressor [69,70].

## 11. Conclusions

Our understanding of glaucoma as a complex neuroinflammatory and neurodegenerative disease that requires treatment modalities beyond IOP lowering is evolving. A number of mechanisms link stress, neuroinflammation, and glaucoma. Particularly, glucocorticoid-mediated microglial activation may create a vicious cycle of neuroinflammation and the upregulation of HPA-axis activity, leading to autonomic dysfunction. Specifically, in NTG, this autonomic dysfunction can be associated with an inability to regulate the neurovascular environment, which can be worsened by mitochondrial weathering in the setting of chronic stress. This multifactorial insult to energetically-sensitive RGCs may play a role in glaucoma progression. Notably, vascular inflammation due to endothelial dysfunction from DM has been associated with the pathogenesis of both DR and glaucoma. Additionally, the concept of allostatic load as a framework to link chronic life adversity and neuroendocrine dysregulation has been explored in the setting of neuroinflammation and glaucoma. The neurovascular mechanism and HPA-mediated neuroinflammation described above play integral roles in allostatic load and glaucoma progression. Additionally, chronic psychosocial stress has been shown to influence neuroinflammation via cellular signaling by proinflammatory cytokines. 

These stress-associated mediators of neuroinflammation and neurodegeneration may play a role in explaining the health disparities associated with glaucoma. The SES plays a complex role in glaucoma development and progression, as multi-level factors such as access to care and health literacy have been implicated in the disease outcomes. Allostatic load and chronic stress may serve as a conceptual framework to link the SES to glaucoma progression via the aforementioned neuroinflammation pathways; however, further research is needed to ascertain causality. 

Similarly, racial disparities are significant in glaucoma. Although there has been an increasing effort to recruit African-ancestry patients to research studies, there is still a need for further study into the multifactorial nature of this disparity. Barriers to healthcare access have been implicated, even after a correction for the SES, which are likely driven at least in part by systemic inequalities. Additionally, genome-wide association studies and molecular and tissue analyses have suggested that some individuals of African ancestry may be more susceptible to neuroinflammation and ischemic optic nerve damage. There is a well-described link between race and allostatic load, with emerging interest in studying how frequent experiences with discrimination may affect chronic autonomic dysregulation. The allostatic burden brought on by structural and systemic inequality may be associated with worsening neuroendocrine and neuroinflammatory outcomes in a population that is indicated to be genetically susceptible to these factors already.

This allostatic load hypothesis may serve as a useful approach for implicating chronic stress, social determinants of health, genetics, and neuroinflammation in the established risk of glaucoma development and progression in the African-ancestry population. More research is needed to investigate this link further, but it may serve as an interesting avenue for interdisciplinary collaboration between medicine, public health, and public policy to improve the disparities in glaucoma. The relationships described in this paper are summarized at a high level in the general overview in Figure 1:

In conclusion, there is evidence that stress has a multifactorial relationship with neuroinflammation and glaucoma development. These effects may be seen at the molecular and cellular levels, and they may also contribute to macroscopic disparities in glaucoma outcomes, such as the racial and socioeconomic differences described above. More research is needed into how interventions that target chronic stress and neuroinflammation may improve glaucoma outcomes, both on a microscopic level and on a community and health-policy level. Additionally, further research is needed to create risk models that identify patients who are at a particularly high risk for glaucoma development or progression based on the multi-level factors described in this review. Further investigation regarding how these at-risk patients may benefit from differing management is also an interesting avenue for future study.

## Figures and Tables

**Figure 1 ijms-25-01653-f001:**
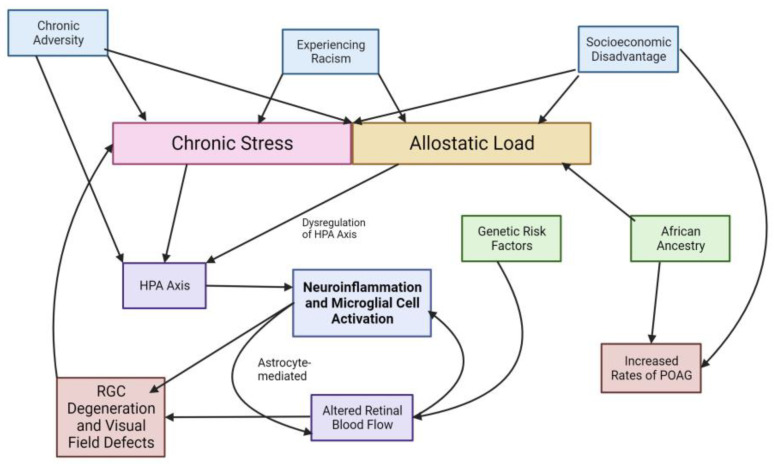
Relationships between multi-level factors and glaucoma.

## Data Availability

Not applicable.

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
