# Peer review of "Stress, Allostatic Load, and Neuroinflammation: Implications for Racial and Socioeconomic Health Disparities in Glaucoma"

_ijms, 2024, doi:10.3390/ijms25031653_

Round 1

Reviewer 1 Report

Comments and Suggestions for Authors

The manuscript entitled "Stress, Allostatic Load, and Neuroinflammation: Implications for Racial and Socioeconomic Health Disparities in Glaucoma" is based on the neuroinflammatory pathways seen in glaucoma and the effects of stress-associated autonomic dysfunction.

The topic is of clinical interest considering the high prevalence of this silent disease, which has a great impact on visual field progression and quality of life. Individuals defined as cortisone responders are well-known risk patients who can show increased IOP when treated with any form of cortisone. The paper nicely shows how endogenous glucocorticoids in response to stress, in addition to racial differences, are important factors to consider when managing patients with glaucoma, ocular hypertension, or have important risk factors in developing glaucoma.

The review has been correctly planned and represents a solid basis for future studies regarding potential novel targets for diagnosis and treatment. It is nicely written and of clinical interest. References are appropriate. The figure is pertinent, and descriptive and assists in describing the mechanisms involved.

The authors should consider adding a flowchart on the management of individuals with risk factors reported in the review, which can help clinicians in managing patients.

Author Response

Thank you for your insightful comments. We have responded to them systematically in the attached file.

Reviewer 2 Report

Comments and Suggestions for Authors

The paper "Stress, Allostatic Load, and Neuroinflammation: Implications for Racial and Socioeconomic Health Disparities in Glaucoma" is an interesting and comprehensive review presenting a novel perspective on the pathogenesis of glaucoma. 

The authors describe the mechanism through which HPA-axis dysregulation can affect neuroinflammation, underscoring the importance of continuous research to identify potential molecular targets for glaucoma therapies. It would also be valuable to elaborate on the mechanism of a potential therapy addressing this pathogenetic pathway in glaucoma development, specifying its potential form of application. Introducing a novel approach and an innovative drug treatment method could represent a significant stride in improving the treatment and prognosis of individuals with glaucoma.

Regarding the claim, "about 40% of the general population has an increased IOP in response to exogenous steroids, compared to nearly 100% of POAG patients. There is also evidence that individuals who demonstrate increased IOP when exposed to exogenous steroids may be at increased risk for POAG in the future (line 130-132)," it is essential to provide the precise reference for this statement and elucidate the research from which these results were derived.

It is necessary to include a section addressing whether there is a distinction in the impact of locally and systemically applied glucocorticoids on neuroinflammation and the pathogenesis of glaucoma.

Additionally, it is essential to cite research on the association between diabetes mellitus and diabetic retinopathy (DR) with glaucoma. This extension should determine whether this connection can be explained through mechanisms such as low-grade inflammation, neuroinflammation, and neurovascular dysregulation. It is known that these mechanisms play a role in the pathogenesis of DR, and there are studies examining the link between DR and glaucoma.

The section addressing the genetic influence on the development of glaucoma requires expansion, incorporating data from pertinent studies. Additionally, it is necessary to list potential genes associated with a predisposition to glaucoma. Furthermore, it should be mentioned if there is research data on the influence of genetic factors on increased sensitivity to exogenous corticosteroids in terms of elevating IOP.

Author Response

Thank you so much for your insightful comments. We have responded to the comments systematically in the attached document.

Round 2

Reviewer 2 Report

Comments and Suggestions for Authors

The authors have responded appropriately to the suggestions and have implemented all the requested changes, resulting in an improved quality of the manuscript. Therefore, the manuscript can be accepted in its current form.